# Proton pump inhibitor use increases the risk of peritonitis in peritoneal dialysis patients

Sayaka Maeda[1], Makoto Yamaguchi[2], Kunihiro Maeda[1], Naoto Kobayashi[1], Naoki Izumi[1], Masaaki Nagai[1], Takaaki Obayashi[1], Wataru Ohashi[3], Takayuki Katsuno[2], Hironobu Nobata[2], Yasuhiko Ito[2]*

1 Department of Nephrology, Narita Memorial Hospital, Toyohashi, Japan, 2 Department of Nephrology and Rheumatology, Aichi Medical University, Nagakute, Japan, 3 Division of Biostatistics, Clinical Research Center, Aichi Medical University, Nagakute, Japan

* yasuito@aichi-med-u.ac.jp

## Abstract

Peritonitis is a major and the most significant complication of peritoneal dialysis (PD). Although some predictors of peritonitis in PD patients are known, the association between proton pump inhibitor (PPI) use and peritonitis has not been characterized. Here, we examined whether PPI use is a risk factor for the development of peritonitis, based on a single-center retrospective analysis of 230 consecutive Japanese PD patients at Narita Memorial Hospital. We assessed the association between PPI use and subsequent first episode of peritonitis using multivariate Cox proportional hazards models, following adjustment for clinically relevant factors. The median follow-up period was 36 months (interquartile range, 19–57 months). In total, 86 patients (37.4%) developed peritonitis. Analysis with multivariate Cox proportional hazards models revealed the following significant predictors of peritonitis: PPI use (adjusted hazard ratio [HR] = 1.72, 95% confidence interval [CI]: 1.11–2.66; P = 0.016) and low serum albumin level (per g/dl adjusted HR = 0.59, 95% CI: 0.39–0.90; P = 0.014). Thus, PPI use was independently associated with PD-related peritonitis. The results suggest that nephrology physicians should exercise caution when prescribing PPIs for PD patients.

## Introduction

Peritonitis is a major and the most significant complication of peritoneal dialysis (PD), which is associated with significant morbidity, catheter loss, transfer to hemodialysis, and permanent membrane damage, and occasionally death [1]. Therefore, it is important to prevent and reduce the risk for developing peritonitis in patients on PD. Important modifiable risk factors, such as recent invasive procedures (colonoscopy, sigmoidoscopy, cystoscopy, hysteroscopy), nasal *Staphylococcus aureus* carriage, and exit-site and/or tunnel infections [2–5] have been identified; in addition to these risk factors, constipation, smoking, domestic pets, obesity, depression, hypokalemia, and hypoalbuminemia have been shown as predictors of peritonitis [6–9].

**Data Availability Statement:** All relevant data are within the paper and its Supporting Information files.

**Funding:** The authors received no specific funding for this work.

**Competing interests:** The authors have declared that no competing interests exist.

Although proton pump inhibitors (PPIs) are among the top 10 most widely used drugs in the world, PPI use has been associated with increased risk of enteric infections, such as *Clostridium difficile* infection and spontaneous bacterial peritonitis (SBP) in cirrhosis patients [10–14]. This is probably due to bacterial overgrowth within the gastrointestinal tract and translocation across the epithelial barrier by usage of acid-suppressive therapy [11,12].

Regarding PD patients, the relationship between PPIs and peritonitis has not been extensively studied. Zhong et al. reported a meta-analysis showing a significant association between H2-receptor antagonist (H2RA) use and "enteric peritonitis"; peritonitis was caused by enteric bacteria, but PPI use was not identified as a risk factor [15]. However, the study sample size was small, which might have led to the underestimation of the influence of PPIs. Therefore, the result remains to be elucidated and should be validated. The aim of the present study was to examine whether PPI use might be a risk factor for peritonitis using a large retrospective PD cohort in Japan.

## Materials and methods

### Study population and data source

The present study included patients aged >20 years undergoing PD as renal replacement therapy between January 1997 and December 2017 at Narita Memorial Hospital. Among the total of 252 consecutive patients, 22 (8.7%) were excluded because of missing data. Finally, 230 PD patients (91.2%) were included (Fig 1).

All data were fully anonymized and that the ethics committee of Narita Memorial Hospital (approval number: 29-12-01) waived the requirement for obtaining informed consent from the patient because of the retrospective nature of this study.

### Data collection

Baseline characteristics at the start of PD, including age, sex, body mass index, laboratory data (hemoglobin, serum albumin, serum potassium, C-reactive protein, estimated glomerular

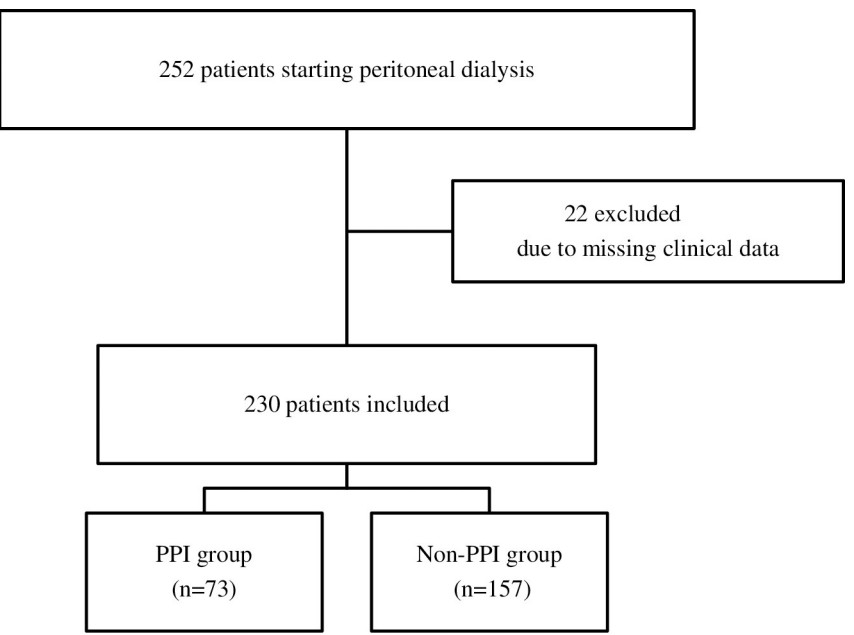

**Fig 1. Flow diagram of the patient selection.**

filtration rate [eGFR]; estimated using the equation recently generated by the Japanese Society of Nephrology: eGFR [mL/min/1.73 m$^2$] = 194 × Scr$^{-1.094}$ × Age$^{-0.287}$ × 0.739 [if female] [16]), urine output per day, and peritoneal transport characteristics (D/P creatinine at 240 minutes during peritoneal equilibration test), hypertension, diabetes mellitus, previous atherothrombotic events (coronary heart disease, thromboembolic stroke, aortic aneurism and/or peripheral vascular disease requiring intervention or hospital admission), cause of kidney disease (diabetic nephropathy, glomerulonephritis, and renal sclerosis), domestic pet, smoking, constipation (defined as a state of using laxative), and the usage of any immunosuppression (previous and ongoing) were collected retrospectively from the medical records.

Tenckhoff catheters (Hayashidera Co. Ltd., Ishikawa, Japan) were placed using a sterile surgical technique and conventional PD solutions (Dianeal-N. 1.5% or 2.5% dextrose, and icodextrin; Baxter Healthcare, Tokyo, Japan), and Y-sets and twin-bag systems were utilized in all PD patients. Patients and their caregivers underwent a standard training program after catheterization.

The characteristics of peritonitis were evaluated in terms of the organisms isolated from the PD effluent. PD effluent was obtained aseptically and inoculated into blood culture bottles. Identification of isolates was performed.

PD-related peritonitis was diagnosed if at least 2 of the following diagnostic criteria were met: (1) abdominal pain or cloudy PD effluent; (2) leukocytosis in the peritoneal fluid effluent (white blood cells >100/mm$^3$, with at least 50% polymorphonuclear neutrophils); or (3) a positive Gram stain or positive culture from PD effluent [17].

The anonymous data set is shown in S1 Table.

## Exposure and outcomes

The main exposure of interest was the PPI during the follow-up period. The primary outcome was the first episode of peritonitis caused by any organism (Gram-positive and -negative organisms and fungi), including culture-negative peritonitis.

Patients who took any PPI for at least 1 week continuously were included in the PPI group, whereas the remaining patients were categorized into the non-PPI group, as previously reported [14]. In those who developed peritonitis in the PPI group, only the patients who took PPI before developing peritonitis were included in the PPI group. We also obtained information regarding the use of H2RA, which was defined as at least 1 week continuously of prescribed H2RA. PPIs included the following drugs: omeprazole, esomeprazole, lansoprazole, rabeprazole, or vonoprazan. H2RA included the following drugs: cimetidine, ranitidine, or famotidine.

Patients were followed up until the first episode of peritonitis or other censoring events, including loss to follow-up, death (cardiovascular disease, malignancy, infection, and others), end of PD, or end of the follow-up for this study, whichever happened to be earlier. We also obtained other outcomes, including recurrence of peritonitis (≧2 episodes of peritonitis), PD withdrawal and its cause (peritonitis, peritoneal dysfunction, impairment of activities of daily living (ADLs), and kidney transplantation), and encapsulating peritoneal sclerosis (EPS).

## Statistical analysis

Differences in clinical characteristics and outcomes between the PPI and non-PPI groups were compared by using the Wilcoxon rank-sum test or Fisher's exact test. To evaluate predictors of the first episode of peritonitis, univariate and multivariate Cox proportional hazards (CPH) models were constructed, including clinically relevant factors as previously reported [2–8].

The proportional hazards assumption for covariates was tested using scaled Schoenfeld residuals. For continuous variables, the Wilcoxon rank-sum test was performed to assess the significance of intergroup differences. Categorical variables were expressed as percentages and compared using Fisher's exact test. The cumulative probability of the development of the first episode of peritonitis was calculated using the Kaplan-Meier method and log-rank test. The level of statistical significance was set at $P<0.05$. All statistical analyses were performed using JMP version 14.0.0 (SAS Institute, Cary, NC, USA).

## Results

### Study participants and clinical characteristics

The present study included 230 PD patients, with 73 (31.7%) patients in the PPI group and 157 (68.3%) patients in the non-PPI group. The baseline characteristics of the two groups are summarized in Table 1. The PPI group had a higher proportion of previous atherothrombotic events (32.9% vs. 19.8%, P = 0.030) than the non-PPI group. The other factors at baseline were not significantly different between the two groups.

### Outcome data

**PD retrieval.**
**Peritonitis incidence**
During the follow-up period (median, 36 months; interquartile range, 19–57 months), 86 patients (37.4%) developed at least one episode of peritonitis. Forty-one (56.2%) and 45 (28.7%) patients in the PPI and non-PPI groups, respectively, developed peritonitis at least once (P<0.001; Table 2). Among the total of 41 patients who developed peritonitis in the PPI group, 36 (87.8%) developed peritonitis during the period of PPI use. The remaining 5 (12.2%) patients developed peritonitis after discontinuation of PPI, but these patients had taken H2RA instead of PPI. The proportion of recurrent episodes of peritonitis (≥2 episodes) was higher in the PPI group than in the non-PPI group (19 [26.0%] patients vs 24 [15.3%] patients, P = 0.041). The incidence of peritonitis was 0.30 and 0.18 person-year in the PPI and non-PPI groups, respectively. The cumulative probabilities of the first episode of peritonitis at 1, 3, and 5 years were 0.22, 0.43, and 0.63, respectively, in the PPI group and 0.14, 0.22, and 0.39, respectively, in the non-PPI group, indicating that the PPI group had a higher risk for developing peritonitis than did the non-PPI group (log-rank test: P = 0.003; Fig 2). Furthermore, the cumulative probabilities of the first episode of peritonitis in those who took PPI at the onset of peritonitis are shown in S1 Fig. The cumulative probabilities of peritonitis at 1, 3, and 5 years were 0.19, 0.69, and 0.94, respectively.

**Predictors of peritonitis**
In the univariate models, low serum albumin level and PPI use were significantly associated with overall peritonitis. Multivariate adjustment for clinically relevant factors attenuated the association between low serum albumin (per 1 g/dl adjusted HR = 0.59, 95% CI): 0.39–0.90; P = 0.014), PPI use (HR, 1.73; 95% CI, 1.12–2.68; P = 0.013), and peritonitis (Table 3). H2RA use was not identified as a risk factor for peritonitis in the univariate and multivariate models.

**PD withdrawal.** PD withdrawal occurred in 57 (78.1%) and 122 (77.7%) patients in the PPI and non-PPI groups, respectively. Among the causes of PD withdrawal, peritonitis occurred in 11 (19.3%) and 17 (13.9%) patients, peritoneal dysfunction in 15 (26.3%) and 40 (32.8%), impairment in ADLs in 3 (5.3%) and 12 (9.8%), and renal transplantation in 3 (5.3%) and 2 (1.6%) patients of the PPI and non-PPI groups, respectively (P = 0.664).

**Other outcomes.** During the observation period, 10 (13.7%) and 19 (12.1%) patients in the PPI and non-PPI groups, respectively, had mortality events of all causes, indicating that the cause of death was not different between the two groups (P = 0.434; Table 2).

**Table 1. Comparison of baseline characteristics between the proton pump inhibitor (PPI) (n = 73) and non-PPI (n = 157) groups.**

| | PPI group (n = 73) | Non-PPI group (n = 157) | *P* value |
|---|---|---|---|
| Age (year) | 64 (54–72) | 64 (56–77) | 0.649 |
| Male (N (%)) | 53 (72.6) | 112 (71.3) | 0.823 |
| Body mass index (kg/m$^2$) | 22.2 (19.7–24.1) | 22.3 (20.0–24.7) | 0.587 |
| Hemoglobin (g/dL) | 9.9 (9.0–11.3) | 9.9 (9.0–11.0) | 0.985 |
| Serum albumin (g/L) | 3.4 (2.9–3.8) | 3.5 (3.1–3.9) | 0.193 |
| Serum potassium (mEq/L) | 4.1 (3.5–4.8) | 4.2 (3.6–4.7) | 0.257 |
| CRP (mg/dL) | 0.3 (0.1–2.6) | 0.2 (0.1–0.7) | 0.374 |
| eGFR (mL/m/1.73 m$^2$) | 7.6 (5.7–9.7) | 6.9 (5.7–8.7) | 0.129 |
| Urine output (ml/day) | 1040 (700–1400) | 1000 (800–1315) | 0.726 |
| D/P creatinine | 0.70 (0.58–0.79) | 0.67 (0.58–0.77) | 0.889 |
| Hypertension | 64 (87.7) | 125 (79.6) | 0.137 |
| Diabetes mellitus | 42 (57.5) | 71 (45.2) | 0.082 |
| Previous atherothrombotic event | 24 (32.9) | 31 (19.8) | 0.030 |
| Usage of immunosuppression | 3 (4.1) | 15 (9.6) | 0.152 |
| Use of H2RA | 10 (13.7) | 26 (16.6) | 0.698 |
| Cause of kidney disease | | | 0.442 |
| Diabetic nephropathy | 43 (58.9) | 78 (49.7) | |
| Glomerulonephritis | 20 (27.4) | 43 (27.4) | |
| Renal sclerosis | 2 (2.7) | 13 (8.3) | |
| Others | 8 (11.0) | 23 (14.6) | |
| Domestic pet | 16 (21.9) | 32 (20.8) | 0.844 |
| Smokers (current/ex-) | 16 (21.9) | 26 (16.6) | 0.683 |
| Constipation (use of laxative) | 51 (69.9) | 114 (72.6) | 0.667 |

Median (interquartile range), categorical values are expressed as number (proportion).

Conversion factors for units: SCr in mg/dL to μmol/L, × 88.4; eGFR (mL/min/1.73 m$^2$) = 194 × Scr$^{-1.094}$ × Age$^{-0.287}$ × 0.739 (if female)

Abbreviations: eGFR, estimated glomerular filtration rate; H2RA, H2-receptor antagonist; CRP, C-reactive protein; D/P, dialysate/plasma ratio

### Organisms that caused peritonitis in the two groups

Table 4 shows the distributions of the causative pathogens of PD-related peritonitis in both the PPI and non-PPI groups. Among the 86 patients with the first episode of peritonitis, 41 (47.7%) patients were in the PPI group and 45 (52.3%) patients were in the non-PPI group. The organisms isolated from the PD effluent were not different between the PPI and non-PPI groups (P = 0.808).

Gram-positive bacteria were the most common pathogens, accounting for 33.7% of the bacteriologic cultures (n = 29; 15 [36.6%] in the PPI group and 14 [31.1%] in the non-PPI group). Gram-negative bacteria accounted for 14.0% of the bacteriologic cultures (n = 12; 7 [17.1%] in the PPI group and 5 [11.1%] in the non-PPI group). Culture-negative peritonitis was observed in 29 (33.7%) patients, including 14 (34.2%) patients from the PPI group and 15 (33.3%) from the non-PPI group.

### Discussion

In this retrospective single center cohort of 230 consecutive Japanese PD patients, we evaluated the association of PPI use and peritonitis. We found that PPI use was associated with increased risk of peritonitis.

**Table 2. Comparison of outcome between the proton pump inhibitor (PPI) (n = 73) and non-PPI (n = 157) groups.**

| | PPI group (n = 73) | Non-PPI group (n = 157) | P value |
|---|---|---|---|
| **Outcomes** | | | |
| **PD retrieval** | | | |
| **Peritonitis incidence** | | | |
| Peritonitis (at least one episode) | 41 (56.2) | 45 (28.7) | <0.001 |
| Peritonitis (≥2 episode) | 19 (26.0) | 24 (15.3) | 0.041 |
| **PD withdrawal** | 57 (78.1) | 122 (77.7) | 0.664 |
| Peritonitis | 11 (19.3) | 17 (13.9) | |
| Peritoneal dysfunction | 15 (26.3) | 40 (32.8) | |
| ADL impairment | 3 (5.3) | 12 (9.8) | |
| Renal transplantation | 3 (5.3) | 2 (1.6) | |
| Others | 10 (17.5) | 20 (16.3) | |
| EPS | 2 (2.7) | 1 (0.6) | 0.190 |
| **Other outcomes** | | | |
| Death | 10 (13.7) | 19 (12.1) | 0.434 |
| Cardiovascular disease | 5 (50.0) | 10 (52.6) | |
| Malignancy | 0 (0) | 2 (10.5) | |
| Infection | 5 (50.0) | 3 (15.8) | |
| Others | 0 (0) | 4 (21.1) | |
| Observation period (months) | 44 (22–60) | 34 (18–56) | 0.147 |

Median (interquartile range), categorical values are expressed as number (proportion).

Abbreviations: EPS, encapsulating peritoneal sclerosis; CVD, cardiovascular disease, PD, peritoneal dialysis; ADL, activity of daily living

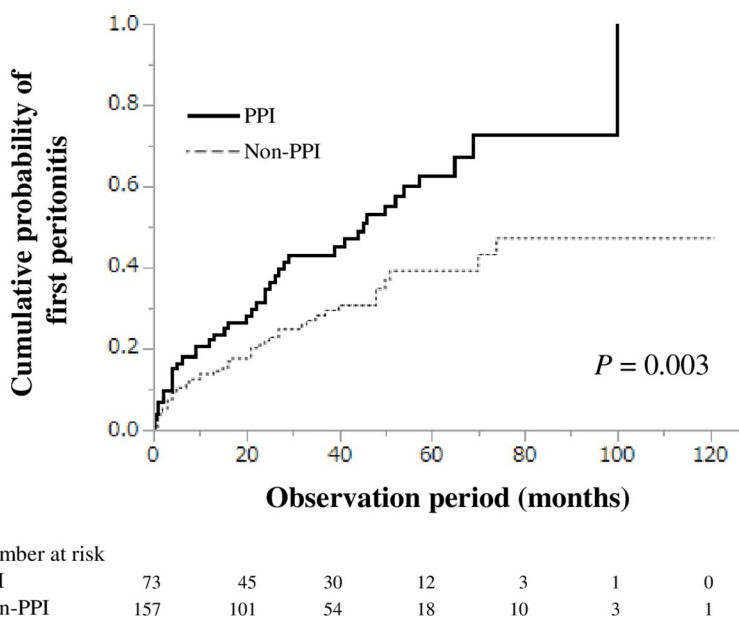

Number at risk

| | | | | | | | |
|---|---|---|---|---|---|---|---|
| PPI | 73 | 45 | 30 | 12 | 3 | 1 | 0 |
| Non-PPI | 157 | 101 | 54 | 18 | 10 | 3 | 1 |

**Fig 2. Cumulative probability of the first episode of peritonitis.**

**Table 3. Predictors of peritonitis.**

| | Univariate model | | Multivariate model | |
|---|---|---|---|---|
| | HR (95% CI) | *P* value | HR (95% CI) | *P* value |
| Age (per 10 years) | 1.14 (0.95–1.37) | 0.141 | 1.03 (0.85–1.26) | 0.758 |
| Male (vs. female) | 1.34 (0.83–2.17) | 0.227 | 1.24 (0.76–2.02) | 0.388 |
| Diabetes | 1.80 (1.17–2.77) | 0.007 | 1.44 (0.92–2.28) | 0.114 |
| Previous atherothrombotic event | 1.30 (0.80–2.12) | 0.296 | 0.98 (0.59–1.64) | 0.948 |
| Usage of immunosuppression | 0.88 (0.40–1.90) | 0.739 | 0.87 (0.38–1.98) | 0.738 |
| Serum albumin (per 1.0 g/dL) | 0.51 (0.35–0.75) | <0.001 | 0.59 (0.39–0.90) | 0.014 |
| PPI use | 1.90 (1.24–2.90) | 0.003 | 1.72 (1.11–2.66) | 0.016 |
| H2RA use | 1.11 (0.65–1.92) | 0.699 | 1.13 (0.63–2.01) | 0.682 |

HR, hazard ratio; CI, confidence interval

PPI, proton pump inhibitor; H2RA, H2-receptor antagonist

Data are the HR, 95% CI, and *P* value from Cox proportional hazard regression analyses.

Adjusted for clinical characteristics including age, sex, diabetes, previous atherothrombotic event, usage of immunosuppression, serum albumin level, PPI use, and H2RA use.

PPIs are used worldwide, and the ratio of PPI usage was higher in PD patients because of various gastrointestinal conditions, such as gastroesophageal reflux disease, peptic ulcer, and ulcer prophylaxis for anti-platelet therapy [2].

In cirrhosis patients, the clinical risk of PPI use in developing SBP was reported in several meta-analyses, including well-designed large cohort studies [10–14]. The potential mechanisms for PPIs to increase the risk of SBP have been reported in several previous studies [17–21]. Some in vitro studies showed that PPIs affect the inflammatory cells, including lymphocytes, neutrophils, or natural killer cells, directly. Other studies have indicated that PPIs inhibit the oxidative burst in human neutrophils and disturb the response by dendritic cells to microbial ligands [18,19]. These impaired functions of the immune cells might promote the translocation of intestinal bacteria, thereby leading to peritonitis. All of these effects may cause

**Table 4. Comparison of isolated organism of peritonitis between the proton pump inhibitor (PPI) (n = 41) and non-PPI (n = 45) groups.**

| | PPI group (n = 41) | Non-PPI group (n = 45) | *P* value |
|---|---|---|---|
| **Organisms** | | | 0.808 |
| Gram-positive | 15 (36.6) | 14 (31.1) | |
| Coagulase-negative *Staphylococcus* | 4 | 4 | |
| *Staphylococcus aureus* | 5 | 6 | |
| *Streptococcus* species | 5 | 3 | |
| *Enterococcus* species | 1 | 1 | |
| Gram-negative | 7 (17.1) | 5 (11.1) | |
| *Pseudomonas* species | 2 | 0 | |
| *Klebsiella pneumoniae* | 0 | 2 | |
| *Acinetobacter baumannii* | 0 | 1 | |
| Others | 5 | 2 | |
| Fungi | 1 (2.4) | 1 (2.2) | |
| Culture-negative | 14 (34.2) | 15 (33.3) | |
| Others | 4 (9.8) | 10 (22.2) | |

Median (interquartile range), categorical values are expressed as number (proportion).

changes in the natural gut microbial environment, which subsequently leads to increased bacterial colonization in the gastrointestinal tract [20,21]. Overall, PPIs may predispose patients to bacterial overgrowth within the gastrointestinal tract and translocation across the impaired epithelial barrier [21], which increases the risk for peritonitis development.

Meanwhile, in PD patients, few studies have reported the association between PPIs and PD-related peritonitis [15,22–24], and no previous studies have shown the significant association between PPIs and PD-related peritonitis. Thus far, only one recent meta-analysis consisting of 6 observational studies involving 378 PD patients has evaluated the relationship between PPI, H2RA, and enteric peritonitis [15]. Although the results showed that H2RA use in PD patients was associated with an increased risk of enteric peritonitis (odds ratio [OR] = 1.27; 95% CI: 1.02-1.57), PPI use was not identified as a risk factor (OR = 1.13; 95% CI: 0.72-1.77); the results should be interpreted cautiously in the following points.

As for the methodological point, the meta-analysis included a small number of patients, which may have underestimated the influence of PPI on peritonitis development. Furthermore, although Pérez [22], included in the meta-analysis, showed that H2RA use, at the time of starting PD, was associated with peritonitis, PPIs did not show a significant association with peritonitis in their Cox proportional hazard model. However, when taking into account the data on H2RA use during the follow-up period in the time-dependent Cox proportional hazard models, a significant association between H2RA and peritonitis was not found; therefore, the result should be interpreted cautiously.

As for the point on the pathological mechanism of the drug for peritonitis, in the above-mentioned meta-analysis, the difference in the influence for peritonitis between PPI and H2RA was considered to be the difference in the pharmacokinetics between PPI and H2RA. Namely, the clearance of PPI showed no significant pharmacokinetics between patients with renal failure and healthy volunteers [25]. Meanwhile, the metabolism of ranitidine, a H2RA drug, was reduced in PD patients [26], resulting in more lasting effects of H2RA in these patients, which may increase the risk of developing peritonitis. However, the dose of ranitidine examined in the study [26] was higher than the current recommended dose for PD patients; therefore, the result could not be applied. In a different point of view, most of the previous studies in PD patients had evaluated the association between PPI or H2RA and "enteric" peritonitis, which was caused by the enteric organisms [15,22,23,26]. However, the definition of "enteric" organisms was different in each study, namely, enteric bacteria included in each study was different. Furthermore, generally, previous studies did not include peritonitis caused by *Streptococcus* or *Staphylococcus* for "enteric" organisms. However, some studies showed that *Streptococcus* and *Staphylococcus* increased significantly in the gut of PPI users [21,27–29]. Therefore, the peritonitis caused by *Streptococcus* and *Staphylococcus* should be evaluated in addition to previously defined enteric organisms to examine the relationship between PPI and peritonitis.

Contrary to the previous studies, the present study showed that H2RA was not a risk factor for peritonitis development. The reason was unclear, but it might be that bacterial colonization of the small intestine and bacterial overgrowth might occur more easily in subjects using PPIs, because PPIs are associated with stronger acid suppression than H2RA even in PD patients with kidney dysfunction, as previously reported in patients with normal kidney function [28, 29]. Given that the effect of both PPI and H2RA for gut microbial environment was not directly compared, it is unknown which drug has a stronger influence on the intestinal environment in PD patients. Therefore, further studies should be undertaken in the future to clarify the mechanism.

Interestingly, as for the pathophysiological mechanism of PD-related peritonitis, the environment of gut microbiomes was different among races, which might be due to the differences

in genes, eating habits, living environments, and metabolic levels [23,30]. The difference in gut microbiome might have a different influence on PPIs, resulting in different risks for peritonitis development in each race. Therefore, in the future, results should be validated in various populations through well-designed studies.

This study also showed that a lower serum albumin level was a significant risk factor for peritonitis, as previously reported [8]. Hypoalbuminemia in patients with renal failure is multifactorial and may result from malnutrition or inflammation, and it is possible that the increased risk for peritonitis in hypoalbuminemic patients relates to an underlying inflammatory state [8]. In the future, it should be assessed whether hypoalbuminemia is a true predictor or simply the effect of inflammation, by analysis of another cohort involving patients with various clinical characteristics; however, the present study suggests that patients with hypoalbuminemia should be carefully managed for peritonitis.

This study has several limitations. First, the retrospective nature of this design, confounded by indication of PPI, was not fully adjusted. Furthermore, we did not obtain information regarding the reason behind the administration of PPI or H2RA. Second, medication compliance with PPI was unadjusted, which may have resulted in a potential bias. Third, information on the dosage of PPI and H2RA treatment was lacking; hence, the intensity of anti-gastric acid effect could not be precisely assessed. Fourth, there was a lack of information regarding exit-site infection, which was identified as a risk factor for peritonitis [31]; therefore, these factors should be assessed in future studies. Fifth, because of the retrospective nature of the present study, we could not determine the basis for starting PPI treatment, and therefore could not establish a causal link between PPI treatment and peritonitis.

Our study has 2 advantages. First, our research is among the largest retrospective cohort studies with a long follow-up period. Second, we were able to assess the relationship between intensive exposure to PPI and peritonitis, which suggested a significant relationship between PPI treatment and peritonitis.

## Conclusion

PPI use was independently associated with PD-related peritonitis. The results suggest that nephrologists should pay attention when prescribing PPI in PD patients.

## Supporting information

**S1 Fig. Cumulative probability of the first episode of peritonitis in those who took PPI at the onset of peritonitis.**
(TIF)

**S1 Table. Anonymous dataset of PD patients.**
(XLSX)

## Acknowledgments

We are grateful to K. Nakabayashi for helping in the data collection and providing useful suggestions.

## Author Contributions

**Data curation:** Sayaka Maeda, Kunihiro Maeda, Naoto Kobayashi, Naoki Izumi, Masaaki Nagai, Takaaki Obayashi.

**Formal analysis:** Makoto Yamaguchi, Wataru Ohashi.

**Methodology:** Takayuki Katsuno.

**Project administration:** Takaaki Obayashi.

**Supervision:** Hironobu Nobata, Yasuhiko Ito.

**Writing – original draft:** Sayaka Maeda, Makoto Yamaguchi, Yasuhiko Ito.

**Writing – review & editing:** Makoto Yamaguchi, Yasuhiko Ito.

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
