## [Decision Letter · Decision Letter 0]

9 Sep 2019

PONE-D-19-22245

Proton pump inhibitor use increases the risk of peritonitis in peritoneal dialysis patients

PLOS ONE

Dear Dr. Ito,

Thank you for submitting your manuscript to PLOS ONE. After careful consideration, we feel that it has merit but does not fully meet PLOS ONE’s publication criteria as it currently stands. Therefore, we invite you to submit a revised version of the manuscript that addresses the points raised during the review process.

As noted in comments by the referee, please specify the issue.

We would appreciate receiving your revised manuscript by Oct 24 2019 11:59PM. To enhance the reproducibility of your results, we recommend that if applicable you deposit your laboratory protocols in protocols.io, where a protocol can be assigned its own identifier (DOI) such that it can be cited independently in the future. For instructions see: http://journals.plos.org/plosone/s/submission-guidelines#loc-laboratory-protocols

We look forward to receiving your revised manuscript.

Kind regards,

Hideharu Abe, M.D., Ph. D.

Academic Editor

PLOS ONE

Journal Requirements:

1. In the ethics statement in the manuscript and in the online submission form, please provide additional information about the patient records used in your retrospective study. Specifically, please ensure that you have discussed whether all data/tissue samples  were fully anonymized before you accessed them and/or whether the IRB or ethics committee waived the requirement for informed consent. If patients provided informed written consent to have data from their medical records used in research, please include this information.

We also ask that you please provide the full name of the institutional review board(s) or the ethics committees that reviewed and approved this study.

2. We noticed minor instances of text overlap with the following previous publication(s), which need to be addressed:

https://bmcinfectdis.biomedcentral.com/articles/10.1186/s12879-019-4300-0

The text that needs to be addressed involves the abstract.

 In your revision please ensure you cite all your sources (including your own works), and quote or rephrase any duplicated text outside the methods section. Further consideration is dependent on these concerns being addressed.

Reviewers' comments:

Reviewer's Responses to Questions

**Comments to the Author**

1. Is the manuscript technically sound, and do the data support the conclusions?

Reviewer #1: Yes

Reviewer #2: Partly

Reviewer #3: Yes

Reviewer #4: Partly

2. Has the statistical analysis been performed appropriately and rigorously? 

Reviewer #1: Yes

Reviewer #2: Yes

Reviewer #3: Yes

Reviewer #4: Yes

3. Have the authors made all data underlying the findings in their manuscript fully available?

Reviewer #1: Yes

Reviewer #2: Yes

Reviewer #3: Yes

Reviewer #4: Yes

4. Is the manuscript presented in an intelligible fashion and written in standard English?

Reviewer #1: Yes

Reviewer #2: Yes

Reviewer #3: Yes

Reviewer #4: Yes

5. Review Comments to the Author

Reviewer #1: Maeda S et al explored the association between proton pump inhibitor use and peritonitis using single-center retrospective cohort in their hospital. As a total of 230 peritoneal dialysis patients in the past 20 years have been enrolled, this study is well designed as retrospective cohort study. However I think this work is suitable for being published in PLOS ONE, it cannot be accepted in its original form. I have a several queries such as unclear causal association between PPI use and the occurrence of peritonitis.

1) The authors define PPI group as ` The patients who took PPI at least 1 week. '. I think it does not include the information about the timing when PPI was taken.

2) Furthermore, this definition does not clarify whether peritonitis was developed within the period the patients took PPI.

3) The authors assume intestinal flora change and bacterial translocation due to oral PPI as the mechanism of peritonitis development. However, logical consistency is insufficient if peritonitis had not occurred within the period when PPI had been prescribed.

The authors should clarify whether the patients were suffered from peritonitis during the period of taking PPI.

Reviewer #2: Dr. Maeda S et al. reported the association between proton pump inhibitor (PPI) use and the risk of peritonitis in peritoneal dialysis patients in the retrospective single center cohort. In general, the work is interesting, although there are several deficiencies that need to be addressed before it would be suitable for publication, including:

1. The authors had defined patients who took any PPI for at least 1 week as the PPI group. How many patients had taken any PPI at onset of peritonitis? The authors should show the Kaplan-Meier curve of patients who took any PPI at onset of peritonitis.

2. The authors should describe the usage of previous/ongoing immunosuppression in Table 1.

3. The observational, nonrandomized design does not permit to establish a causality link between treatment with PPI and outcomes. The authors should well describe the possibility that PPI use may just be a confounding factor for other, unknown variables.in limitation.

Reviewer #3: In this manuscript, the authors evaluated the association of PPI use and peritonitis, and they found that PPI use was associated with increased risk of peritonitis. This paper must be the first one showing the risk of PPI for peritonitis as the authors say and might be very important information in clinical practice for the PD patients, however there seems to be several points should be revised.

1. The authors defined the PPI group as the patients who took any PPI for at least 1 week and did not define the timing of taking the medicine. Please explain why the authors defined the period of prescription as 1 week and if you have some articles you used as reference, please specify them.

2. Patients taking steroid or immunosuppressant are vulnerable to infection. Please describe the number of patients taking steroid or immunosuppressant, if you have data.

3. The reason why the association between H2-receptor antagonist (H2RA) use and peritonitis was not significant might be due to the small sample size. Please clarify the number of the patients taking H2RA in Table 1.

Reviewer #4: In the manuscript entitled “Proton pump inhibitor use increases the risk of peritonitis in peritoneal dialysis patients”, authors conducted a retrospective PD patient study to assess the risk of using PPI against peritonitis. The study is well designed and provides good recommendation for the nephrologists. There are some issues that need to be addressed before publishing.

1) It is not clear where the H2RA patient cohort is coming from? Please make it clear and include the data in Figure 1

2) In the table 2 make it clear that Peritonitis, peritoneal disfunction etc are the reasons for PD retrieval. Because of the way the table is structured, it took some effort to understand. Make some font changes, or tab difference to make it easy to follow

3) In this study, baseline data from the patients show that serum albumin from PPI and non-PPI groups are comparable. It is known that inflammation can cause hypoalbuminemia. In other studies where they predicted serum albumin as risk factor, the base serum level at admission was lower. In the current study, it is not clear the low serum albumin is really a predictor or the effect of inflammation, please comment and clarify.

4) A minimum treatment of 1 week with PPI was included in the PPI group. Was the treatment continuous, or intermittent?

5) Authors often compared with another meta analysis and commented that the current work has more sample number. It is not true. Please correct the facts contextually.

6. PLOS authors have the option to publish the peer review history of their article (what does this mean?). If published, this will include your full peer review and any attached files.

Reviewer #1: No

Reviewer #2: No

Reviewer #3: No

Reviewer #4: No

---

## [Author Response · Author response to Decision Letter 0]

3 Oct 2019

5. Review Comments to the Author

Reviewer #1: Maeda S et al explored the association between proton pump inhibitor use and peritonitis using single-center retrospective cohort in their hospital. As a total of 230 peritoneal dialysis patients in the past 20 years have been enrolled, this study is well designed as retrospective cohort study. However I think this work is suitable for being published in PLOS ONE, it cannot be accepted in its original form. I have a several queries such as unclear causal association between PPI use and the occurrence of peritonitis.

1) The authors define PPI group as ` The patients who took PPI at least 1 week. '. I think it does not include the information about the timing when PPI was taken.

Patients who took any PPI for at least 1 week were included in the PPI group, whereas the remaining patients were categorized into the non-PPI group, as previously reported [14]. In those who developed peritonitis in the PPI group, only the patients who took PPI before developing peritonitis were included in the PPI group. 

We modified the sentence of the 2nd paragraph in the “Exposure and outcomes” section as follows.

“Patients who took any PPI for at least 1 week continuously were included in the PPI group, whereas the remaining patients were categorized into the non-PPI group, as previously reported [14]. In those who developed peritonitis in the PPI group, only the patients who took PPI before developing peritonitis were included in the PPI group. Patients who took any H2RA for at least 1 week continuously constituted the H2RA group, similar to the definition of the “PPI group”. PPIs included the following drugs: omeprazole, esomeprazole, lansoprazole, rabeprazole, or vonoprazan. H2RA included the following drugs: cimetidine, ranitidine, or famotidine.”

2) Furthermore, this definition does not clarify whether peritonitis was developed within the period the patients took PPI.

Among the total of 41 patients who developed peritonitis in the PPI group, 36 (87.8%) patients developed peritonitis within the period of PPI use. The remaining 5 (12.2%) patients developed peritonitis after stopping PPI, but these patients took H2RA instead of PPI. This suggests that the continued anti-acid effect might predispose the patients to develop peritonitis. 

3) The authors assume intestinal flora change and bacterial translocation due to oral PPI as the mechanism of peritonitis development. However, logical consistency is insufficient if peritonitis had not occurred within the period when PPI had been prescribed.

The authors should clarify whether the patients were suffered from peritonitis during the period of taking PPI.

Among the total of 41 patients who developed peritonitis in the PPI group, only 5 (12.2%) patients developed peritonitis after stopping PPI. However, these patients took H2RA after PPI stopping; therefore, it might be the continued effect of anti-gastric acid suppression by H2RA, which led to intestinal flora changes and bacterial translocation after stopping PPI.

We inserted the following sentence in the first paragraph of the “Peritonitis incidence” subsection under the “Outcome data” section

“Among the total of 41 patients who developed peritonitis in the PPI group, 36 (87.8%) patients developed peritonitis within the period of PPI use. The remaining 5 (12.2%) patients developed peritonitis after stopping PPI, but these patients had taken H2RA instead of PPI.“

Reviewer #2: Dr. Maeda S et al. reported the association between proton pump inhibitor (PPI) use and the risk of peritonitis in peritoneal dialysis patients in the retrospective single center cohort. In general, the work is interesting, although there are several deficiencies that need to be addressed before it would be suitable for publication, including:

1. The authors had defined patients who took any PPI for at least 1 week as the PPI group. How many patients had taken any PPI at onset of peritonitis? The authors should show the Kaplan-Meier curve of patients who took any PPI at onset of peritonitis.

Among the total of 41 patients who developed peritonitis in the PPI group, 36 (87.8%) patients developed peritonitis within the period of PPI use. The remaining 5 (12.2%) patients developed peritonitis after stopping PPI, but these patients took H2RA instead of PPI. As your comments, we showed the Kaplan-Meier curve of patients who took any PPI at onset of peritonitis in the supplemental figure.

We inserted the following sentence in the first paragraph of the “Peritonitis incidence” subsection under the “Outcome data” section

 “Furthermore, the cumulative probabilities of the first episode of peritonitis in those who took PPI at onset of peritonitis were also shown (Supplemental Figure 1).”

2. The authors should describe the usage of previous/ongoing immunosuppression in Table 1.

We have no information about the usage of previous/ongoing immunosuppression. Thus, we inserted the following sentence in the limitation paragraph.

“Fourth, information about exit-site infection and the use of previous/ongoing immunosuppressive treatment, which were identified as risk factors for peritonitis, was lacking; therefore, these factors should be assessed in future studies.”

3. The observational, nonrandomized design does not permit to establish a causality link between treatment with PPI and outcomes. The authors should well describe the possibility that PPI use may just be a confounding factor for other, unknown variables.in limitation.

We inserted the following sentence in the limitation.

 “Fifth, because of the retrospective nature of the present study, we could not determine the basis for starting PPI and establish a causality link between PPI and peritonitis.”

Furthermore, we modified the sentence of last paragraph under the “Discussion” section, as follows.

“Second, we were able to assess the relationship between the intensive exposure of PPI and peritonitis, which suggested a significant relationship between PPI and peritonitis.”

Reviewer #3: In this manuscript, the authors evaluated the association of PPI use and peritonitis, and they found that PPI use was associated with increased risk of peritonitis. This paper must be the first one showing the risk of PPI for peritonitis as the authors say and might be very important information in clinical practice for the PD patients, however there seems to be several points should be revised.

1. The authors defined the PPI group as the patients who took any PPI for at least 1 week and did not define the timing of taking the medicine. Please explain why the authors defined the period of prescription as 1 week and if you have some articles you used as reference, please specify them.

In the present study, PPI use was defined as patients who took any PPI for at least 1 week, as previously reported [14], showing that the steady maximum mean percentage time of gastric pH >4 was seen after taking PPI for 1week.

2. Patients taking steroid or immunosuppressant are vulnerable to infection. Please describe the number of patients taking steroid or immunosuppressant, if you have data.

We have no data about the usage of previous/ongoing immunosuppression. Thus, we inserted the following sentence in the limitation paragraph.

“Fourth, information about exit-site infection and the use of previous/ongoing immunosuppressive treatment, which were identified as risk factors for peritonitis, was lacking; therefore, these factors should be assessed in future studies.”

3. The reason why the association between H2-receptor antagonist (H2RA) use and peritonitis was not significant might be due to the small sample size. Please clarify the number of the patients taking H2RA in Table 1.

We clarified the number of the patients taking H2RA in Table 1.

Reviewer #4: In the manuscript entitled “Proton pump inhibitor use increases the risk of peritonitis in peritoneal dialysis patients”, authors conducted a retrospective PD patient study to assess the risk of using PPI against peritonitis. The study is well designed and provides good recommendation for the nephrologists. There are some issues that need to be addressed before publishing.

1) It is not clear where the H2RA patient cohort is coming from? Please make it clear and include the data in Figure 1

In the present study, we focused on the relationship between PPI use and peritonitis. Therefore, if the information about H2RA was included in Figure 1, it might be confusing. Then, we presented the data of H2RA in Table 1. Please check and advise whether the modifications made are correct or not.

2) In the table 2 make it clear that Peritonitis, peritoneal disfunction etc are the reasons for PD retrieval. Because of the way the table is structured, it took some effort to understand. Make some font changes, or tab difference to make it easy to follow

We modified the structure of Table 2 as per your recommendation.

3) In this study, baseline data from the patients show that serum albumin from PPI and non-PPI groups are comparable. It is known that inflammation can cause hypoalbuminemia. In other studies where they predicted serum albumin as risk factor, the base serum level at admission was lower. In the current study, it is not clear the low serum albumin is really a predictor or the effect of inflammation, please comment and clarify.

Various factors such as inflammation or malnutrition might lead to hypoalbuminemia. The difference of albumin level between the present and other previous studies suggests that the conditions or underlying diseases of the kidney failure might be different among the studied patients. 

Therefore, our results should be validated in other cohorts. 

We modified the following sentence in the Discussion.

“Although in the future, it should be assessed whether hypoalbuminemia is really a predictor or the effect of inflammation in another cohort including patients with various clinical characteristics, the present study suggests that patients with hypoalbuminemia should be carefully managed for peritonitis.”

4) A minimum treatment of 1 week with PPI was included in the PPI group. Was the treatment continuous, or intermittent?

Patients who took any PPI for at least 1 week continuously were included in the PPI group.

We modified the sentence in the 2nd paragraph of the “Exposure and outcomes” section as follows.

“Patients who took any PPI for at least 1 week continuously were included in the PPI group, whereas the remaining patients were categorized into the non-PPI group, as previously reported [14].”

5) Authors often compared with another meta analysis and commented that the current work has more sample number. It is not true. Please correct the facts contextually. 

We consider that it might be possible that the abovementioned meta-analysis including 378 patients might be too small to evaluate the effect the PPI. Therefore, it should be validated in other larger studies.

---

## [Decision Letter · Decision Letter 1]

24 Oct 2019

Proton pump inhibitor use increases the risk of peritonitis in peritoneal dialysis patients

PONE-D-19-22245R1

Dear Dr. Ito,

We are pleased to inform you that your manuscript has been judged scientifically suitable for publication and will be formally accepted for publication once it complies with all outstanding technical requirements.

With kind regards,

Hideharu Abe, M.D., Ph. D.

Academic Editor

PLOS ONE

Additional Editor Comments (optional):

Reviewers' comments:

Reviewer's Responses to Questions

**Comments to the Author**

1. If the authors have adequately addressed your comments raised in a previous round of review and you feel that this manuscript is now acceptable for publication, you may indicate that here to bypass the “Comments to the Author” section, enter your conflict of interest statement in the “Confidential to Editor” section, and submit your "Accept" recommendation.

Reviewer #1: All comments have been addressed

Reviewer #2: All comments have been addressed

Reviewer #3: All comments have been addressed

Reviewer #4: All comments have been addressed

2. Is the manuscript technically sound, and do the data support the conclusions?

Reviewer #1: Yes

Reviewer #2: (No Response)

Reviewer #3: Yes

Reviewer #4: Yes

3. Has the statistical analysis been performed appropriately and rigorously? 

Reviewer #1: Yes

Reviewer #2: (No Response)

Reviewer #3: Yes

Reviewer #4: Yes

4. Have the authors made all data underlying the findings in their manuscript fully available?

Reviewer #1: Yes

Reviewer #2: (No Response)

Reviewer #3: Yes

Reviewer #4: Yes

5. Is the manuscript presented in an intelligible fashion and written in standard English?

Reviewer #1: Yes

Reviewer #2: (No Response)

Reviewer #3: Yes

Reviewer #4: Yes

6. Review Comments to the Author

Reviewer #1: I believe that the authors have completely revised their manuscript R1. I have no more queries or comments about it.

Reviewer #2: (No Response)

Reviewer #3: The issues I pointed out were addressed appropriately. The reason why H2RA use was not identified as a risk factor of peritonitis might be due to the small number of patients using H2RA (n=36). I think this manuscript is acceptable in this form.

Reviewer #4: The Authors have satisfactorily addressed all the comments and made necessary changes in the manuscript.

7. PLOS authors have the option to publish the peer review history of their article (what does this mean?). If published, this will include your full peer review and any attached files.

Reviewer #1: No

Reviewer #2: No

Reviewer #3: No

Reviewer #4: No

---

## [Editor Report · Acceptance letter]

28 Oct 2019

PONE-D-19-22245R1 

Proton pump inhibitor use increases the risk of peritonitis in peritoneal dialysis patients 

Dear Dr. Ito:

I am pleased to inform you that your manuscript has been deemed suitable for publication in PLOS ONE. Congratulations! Your manuscript is now with our production department. 

With kind regards,

on behalf of

Dr. Hideharu Abe 

Academic Editor

PLOS ONE